# The Role of Helium on Ejecta Production in Copper

**DOI:** 10.3390/ma13061270

**Published:** 2020-03-11

**Authors:** Saryu Fensin, David Jones, Daniel Martinez, Calvin Lear, Jeremy Payton

**Affiliations:** 1MST-8, Los Alamos National Laboratory, Los Alamos, NM 87545, USA; djones@lanl.gov (D.J.); daniel_t@lanl.gov (D.M.); crlear@lanl.gov (C.L.); 2P-23, Los Alamos National Laboratory, Los Alamos, NM 87545, USA; payton@lanl.gov

**Keywords:** high strain rate strength, metals, radiation damage

## Abstract

The effect of helium (He) concentration on ejecta production in OFHC-Copper was investigated using Richtmyer–Meshkov Instability (RMI) experiments. The experiments involved complex samples with periodic surface perturbations machined onto the surface. Each of the four target was implanted with a unique helium concentration that varied from 0 to 4000 appm. The perturbation’s wavelengths were λ≈65μm, and their amplitudes $h_{0}$ were varied to determine the wavenumber (2π/λ) amplitude product $kh_{0}$ at which ejecta production beganfor Cu with and without He. The velocity and mass of the ejecta produced was quantified using Photon Doppler Velocimetry (PDV) and Lithium-Niobate (LN) pins, respectively. Our results show that there was an increase of 30% in the velocity at which the ejecta cloud was traveling in Copper with 4000 appm as compared to its unimplanted counterpart. Our work also shows that there was a finer cloud of ejecta particles that was not detected by the PDV probes but was detected by the early arrival of a “signal” at the LN pins. While the LN pins were not able to successfully quantify the mass produced due to it being in the solid state, they did provide information on timing. Our results show that ejecta was produced for a longer time in the 4000 appm copper.

## 1. Introduction

It is well known that when a shock wave in a material reaches a free surface, it can lead to ejection of particles from the surface [1,2,3]; this process and the launched material is referred to as ejecta. Over the last few decades there has been a body of research to understand the relationship between the total ejected mass and parameters associated with the free surface itself. Previous work has shown that the roughness of the free surface is the single most important parameter that impacts ejection of mass from surfaces [4]. Work by Asay et al. [2] also showed that defects on the surface in the form of cavities and scratches can create a surface roughness which leads to ejecta formation. Other works have also shown that impurities along with grain boundaries and slip bands can affect this process [5]. In addition, the total amount of ejected mass has been linked to the volume of surface defects, shape of the shock wave, yield strength of the material, and the phase of the material on release (solid or liquid). Specifically, Androit et al. [6] studied the effect of surface finish on tantalum (Ta) and Tin (Sn), and showed that the presence of grooves on the surfaces leads to higher ejected mass in comparison to polished surfaces. The specific roughness of the machined grooves had the most influence on total ejected mass, even for materials like Sn that melt on release at low pressures. More recent work by Zellner et al. [7] to study the role of surfaces prepared with different processes and final finishes on ejecta production in aluminum 1100 and Sn also showed a similar sensitivity of total ejected mass and the density distribution of ejected fragments to the final finish of the surface. This has also been confirmed with numerous molecular dynamics simulations that suggest surface roughness to be the determining factor in ejecta production [8]. Although, all the studies consistently agree that for a given surface finish, the maximum amount of ejecta is produced when the material is in liquid phase rather than solid. An additional effect of shock-wave shape—square to Taylor—on ejecta production was also documented, showing higher and constantly increasing ejecta production during unsupported shock generated using Taylor waves in materials like Sn [9]. The authors refer readers to Reference [10] for a thorough review of studies that investigate the role of all these factors on total ejecta production. A common theme in all these studies is the focus of these works on single-phase materials.

Although, there do exist a handful of studies that have attempted to investigate ejecta production in multicomponent materials. Androit et al. [6] investigated the effect of density inhomogeneities by using SnPb with 14 wt % Pb and 38 wt% Pb and showed that under the same shock pressure, the amount of ejected mass significantly increased with increasing density inhomogeneities with addition of Pb even in comparison to Sn. This difference was attributed to the impedance mismatch between pure Sn and SnPb eutectic especially because the microstructure of the SnPb alloy consisted of pure Sn grains included in a eutectic SnPb matrix of higher density. This work was also extended to two CuPb alloys with 15 wt% and 36 wt% Pb. Once again, an increase in ejected mass was observed in comparison to pure copper but was attributed to melting of Pb on release. More recently, the authors also investigated the affect of addition of 1–2 wt% lead to copper on both ejecta production [11]. Our results showed that addition of small amounts of lead causes the formation of background ejected mass that is absent in pure Copper. Our work agrees with the initial observations of Androit et al. [6]. However, studies of this nature remain rare and there are still open questions regarding the importance of material microstructure to ejecta production, especially as it pertains to the presence of inhomogeneities in the material.

In this work, we investigate the effect of helium concentration on ejecta production through the use of Richtmyer–Meshkov Instability experiments. The rest of the paper is organized as follows. The next section discusses the experimental techniques employed in this study. Section 3 discusses the results, followed by a brief conclusion.

## 2. Experimental Methodology

In this section, we outline the materials along with the experimental approach utilized in this investigation.

### 2.1. Material and Sample Design

The experiments used a total of four targets; out of which targets 2, 3, and 4 were implanted with 1000, 2000, and 4000 appm of helium, respectively. One of the targets was not implanted with Helium to serve as a reference. The RMI targets were machined as 2-mm thick by 35-mm diameter disks with five different perturbation regions. The perturbations zones were machined to be 3-mm wide separated by 2-mm flat regions, as shown in Figure 1. The perturbations were nominally sine-wave-like features with a wavelength λ of 65 μm. The amplitudes h for the five regions were varied such that the final finishes spanned the kh (where k = 2Π/λ products of 0.5, 0.6, 0.7, 0.8, and 0.9).

These samples were implanted with helium at the Michigan Ion Beam Laboratory (MIBL) at the University of Michigan in a region that was 25 by 6 mm in size, as shown by the red rectangle in Figure 1. Implantation of each sample was carried out using He${}^{+}$ or He${}^{++}$ ions at seven energies: 0.8, 1.4, 2.0, 2.6, 3.2, 3.8, and 4.4 MeV. These sequential, room-temperature implantations were designed to produce partially overlapping peaks in helium distribution from the sample surface to a depth of about 9 μm. Thermal imaging of the samples implanted at MIBL indicate that beam heating was less than 10 ${}^{\circ }$C, while beam current densities were consistently below 1 μA/cm${}^{2}$. All implantations were performed at room temperature and the target chamber vacuum was maintained below 5 × 10${}^{-7}$ torr during the implantation.

Since we were interested in studying the effect of He on ejecta production in copper, it was essential to verify that the initial microstructure (grain size, texture) for copper was not altered during the implantation process. This was essential to ensure quantitative comparison between data obtained from unimplanted and implanted copper. The change in the overall microstructure was quantified before and after irradiation using electron back-scattered diffraction (EBSD) with a FEI Inspect. The morphology of the implanted helium was investigated through the use of transmission electron microscopy (TEM) with a FEI Tecnai F20 (Hillsboro, OR, USA).

To preserve the reflecting surfaces of the RMI targets, characterization specimens were collected from surrogate copper samples prepared from the same plate and implanted under the same conditions as the targets. Standard metallographic techniques were used to prepare samples for EBSD analysis in a FEI Inspect scanning electron microscope (SEM) (Hillsboro, OR, USA). A final polish of μm Al${}_{2}$O${}_{3}$ was followed by an electrochemical polish in a 2:1 solution of phosphoric acid and water at 1.9 V for 30–40 s. The samples were then lightly etched using a solution of FeCl${}_{3}$ and HCl. Surface preparations were completed prior to any implantations. Post-implantation grain sizes (60 μm) and textures (random) were found to be similar to unimplanted copper [12], as shown in Figure 2.

TEM foils were then extracted from these same surrogates and thinned to electron transparency according to conventional “lift-out” techniques in a FEI Helios 600 focused ion beam (FIB) system. Final polishing was performed to minimize preparation-induced damage artifacts, using 2 kV Ga+ ions in the same FIB. Specimen microstructures were primarily examined using bright-field TEM techniques in a 300 kV FEI Tecnai F20, as shown in Figure 3.

The formation of helium bubbles in these samples is to be expected, given the favorable trapping of excess point defects at dissolved helium atoms and the suppressive effect of helium on the critical radii for cavity growth [13]. The authors acknowledge, however, that helium was not introduced homogeneously by the implantation method used here, instead concentrating in bands around the peak ion ranges of each implantation energy. Implantation profiles for each ion energy were simulated using Stopping and Range of Ions in Matter (SRIM) [14] and are shown in Figure 4. With the low diffusivity of helium in copper near room temperature, this leads to a nonuniform distribution of helium bubbles with depth. Future studies on the effects of helium distribution will incorporate ion-energy-degradation techniques to allow for more uniform implantation of helium at elevated temperature.

### 2.2. Impact Experiments

Plate impact experiments were performed on these targets using the 80-mm bore single-stage light gas-gun in MST-8 at LANL. The copper targets were bonded into a lexan plate, 152.4-mm wide by 127-mm high, and 12.7 mm thick, to allow for mounting and alignment in the gas-gun target tank (Figure 5, left). The impact face of the samples, i.e., the side with no perturbations, was 0.4-mm proud of the lexan plate. To ensure a planar impact between the projectile and the target, a mirror was affixed to the front of the target, and a laser was used to align a spot down and back along the 9.2 m barrel. With this method, the deviation from parallel at impact is typically sub-milli-radian. The flyer-plates were tantalum, 50-mm diameter by 2-mm thick, affixed to the front of a lexan projectile. With a projectile velocity of 1.1 km/s, this generated an approximately 1 μs duration square or flat-topped shock in the sample with a peak stress of 30 GPa.

The diagnostics package was mounted on the back of the lexan plate, such that it could cover the rear of the sample where the perturbations were located. This consisted of a round disc with rows of holes to mount either photon Doppler velocimetry (PDV) probes or lithium niobate (LN) pins. The former are used to measure velocity through a noncontact-laser-based interferometric technique. A narrow linewidth laser is used to illuminate the target, and the reflected light is collected by the probe. If the target surface is moving, this reflected light is Doppler shifted. The collected light is mixed with a reference source, producing a beat frequency that is proportional to the surface velocity. Fourier transform techniques are used to extract the velocity-time history. With a sufficiently high-bandwidth oscilloscope, PDV can resolve velocities from rest to 104 m/s with excellent dynamic range. PDV is able to resolve multiple velocities in the field of view at once, hence it is ideally suited to RMI work, where we want to measure the velocities of the ejecta, bubble, and bulk surface. The probes were supplied by AC photonics, part number 1CL15P020LCD01. These are collimating probes with a spot size of approximately 300 μm at a 20-mm distance, as such, each probe captures multiple wavelengths of the perturbation region it is interrogating.

The LN pins were supplied by Dynasen, part number CA-1136-Li-1. They consisted of a 2.4-mm diameter lithium niobate crystal at the end of a 25.4-mm-long brass tube. Lithium niobate is a piezoelectric material, in that it will output a voltage when strained. This voltage can be used to infer (through a series of assumptions such as conservation of momentum) the mass impinging on the LN crystal, providing an estimate of the mass ejected from the perturbed regions of the sample.

A diagram of the diagnostics package is shown in Figure 6. For each perturbation, there was one PDV probe and one LN pin in both the helium-implanted and -unimplanted regions. There were also two PDV probes on the flat regions, one between kh values of 0.6 and 0.5 and the other between kh values of 0.5 and 0.7, to provide a breakout time and the jump-off free surface velocity. A PDV probe was placed looking through the entire target assembly (magenta circle in Figure 6) to measure the projectile velocity. Similarly, another piezoelectric probe (orange circle, Figure 6) was glued onto the lexan plate to provide a voltage spike at target impact to trigger all of the diagnostics. The alignment of the PDV probes to the perturbations was checked with a red laser, to allow visible confirmation of the spot on the desired region. The standoff height—the distance from the rear of the target to the front face of the LN pins and PDV probes—was measured to be 22 mm for all targets. Images of the rear of the mounted target, the diagnostics package, and the full target assembly mounted in the gas gun are shown in Figure 7.

## 3. Results

The four experiments were performed; three at an impact velocity of 1.1 mm/μs (0, 1000, and 2000 appm) and one at 1.0 mm/μs (4000 appm). The velocity–time histories corresponding to these experiments are shown in Figure 8. This data shows the breakout velocities to be ^~^1.3 km/s for the three samples (0, 1000, and 2000 appm) and 1.2 km/s for the 4000-appm copper.

This difference in shock-breakout velocities is important to note when trying to compare the four experiments to each other. Our results also show that perturbations with kh values of 0.8 and 0.9 went unstable and produced ejecta in all the samples, irrespective of helium concentration. This is consistent with previous work performed by Buttler et al. [15]. Figure 9 shows the velocity–time history corresponding to the perturbation with a kh of 0.9 on the four copper samples. We decided to focus on discussing results from a kh of 0.9 in this paper, as the focus of this work is on the effect of helium on ejecta production rather than the effect of kh on ejecta production.

This data shows the formation of a spike which is consistent with the perturbations inverting and growing. There were only minor variations measured in the peak spike velocity as a function of helium concentration. Specifically, the spike velocity was measured to be 2.48, 2.50, 2.48, 2.31 mm/μs with the helium concentration increasing from 0 to 4000 appm. The fact that there is no change in the velocity of the spike, which is related to the high strain rate strength, suggests that at these high He/Dpa ratios, there is almost no change in the material strength. This is in contrast to observations at low strain rate, where an increase in flow stress is observed with helium implantation, especially due to the formation of a gas-bubble superlattice [16,17]. This contrast could be because, in general, the strengthening in materials with helium is due to the helium acting as obstacles to dislocation motion. However, at high strain rates (in this experiment, the strain rate is 10${}^{7}$/s), dislocations do not have time to actually move and interact with the bubbles. So, the instantaneous yield strength is generally due to the shear modulus and the dislocation density of the material [18]. While altering the helium concentration changes the dislocation density somewhat, we postulate that it is not enough to cause a measurable difference in the strength of the material. It is important to note that the decrease in the spike velocity for 4000 appm could be due to the lower shock-breakout velocity and not due to the actual helium-concentration increase.

In addition to the spike velocity, data from Figure 9 can be integrated to extract position–time history, which when coupled with the standoff distance of the LN probes, can be extrapolated to extract an “expected” timing for the impact of ejecta cloud on the LN pins. The velocity-time data shows that there is a slight change in the velocity at which these ejected particles were traveling as a function of helium concentration. Specifically, the ejecta velocity was measured to be 1.72, 1.755, 1.715, and 1.460 mm/μs for the 0-, 1000-, 2000-, and 4000-appm copper, respectively. These velocities coupled with the LN pin standoff distance of 21.8 mm are used to calculate the timing at which ejecta cloud should arrive at the LN pins. This time was calculated to be 12.7, 12.4, 12.7, and 14.9 μs for the 0-, 1000-, 2000-, and 4000-appm copper, respectively. The increase in the timing for the 4000-appm copper is attributed to the lower shock-breakout velocity.

This timing should be consistent with the timing when the LN pins are activated, unless there is material present that is not measured by the PDV probes due to its extremely small size. Figure 10 shows the voltage–time history for the four shots obtained from the LN pins.

These plots show that the arrival time for ejecta on the pins was actually 10 μs for the 0 appm, 12.5 μs for 1000 appm, and 10.5 μs for 2000 appm copper instead of the times obtained from the PDV data. This is an approximately 3 μs difference in the arrival time of ejecta at the LN pins. We hypothesize that this suggests the presence of a fine, atomic-size ejecta cloud that is traveling at a much higher velocity and is too small to reflect light and be captured by the PDV probes. The most curious result is the arrival of ejecta at the LN pins at 7 μs instead of 14 μs for the 4000-appm copper. This is especially interesting because the shock-breakout velocity was lower for the 4000-appm copper compared to the others, which means the ejecta should have been traveling at a lower velocity. This could be due to the presence of helium in the sample, which might be making the material brittle and leading to the formation of finer ejecta that is then moving at a higher velocity. This is a question that is actively being researched right now. It is important to note that ejecta in this work refers to copper atoms only. The helium atoms are too light to be detected by the LN pins. In addition, with the discrepancy in timing it is interesting to note the differences in the total time that signal is produced on the LN pins. This difference in the total time, especially for the 4000-appm copper, suggests the presence of higher amounts of ejecta in the form of copper atoms, leading to the conclusion that ejecta mass is increased as a function of helium concentration. This is somewhat consistent with our recent results from molecular dynamics simulations that show a large difference in total ejected mass from copper with and without helium [19]. Through our modeling work, we had hypothesized that the presence of helium bubbles can lead to an increased ejected mass due to two reasons: (1) The bubbles alter the shock velocity in their vicinity due to differences in their density as compared to copper. This leads to the formation of a nonplanar shock front that can increase ejecta production. (2) The shock wave can compress the bubble and lead to internal jetting, similar to shape-charge phenomenon. So scientifically, an increase in ejecta mass is plausible and is supported by the increase in the time for which an increase in voltage is observed with increasing Helium concentration.

## 4. Conclusions

The goal of this work was to understand the effect of heterogeneities like helium on ejecta production in metals. OFHC copper was used as a model material for this study. Gas-gun driven RMI experiments, which required special perturbations be machined onto the copper targets, were used to investigate the effect of helium concentration on ejecta production. Four different concentrations of helium—0, 1000, 2000, and 4000 appm—were implanted into the copper sample at the Michigan Ion Beam Laboratory. These samples were then subjected to shock loading using a gas-gun, and diagnostics were used to measure the velocity–time history and mass–time history as the perturbations inverted, grew, and eventually went unstable to produce ejecta. Our results show that kh values of 0.8 and 0.9 produced ejecta in all targets regardless of the presence of helium. Further analysis of the velocity and mass data associated with kh of 0.9 showed (1) an increase in the production of finer ejecta traveling at higher velocities as a function of helium concentration, and (2) longer times associated with a finite voltage, suggesting an increase in mass as a function of Helium concentration. These results are significant and show that heterogeneities like Helium can actually alter important dynamic properties like ejecta production from metals.

## Figures and Tables

**Figure 1 materials-13-01270-f001:**
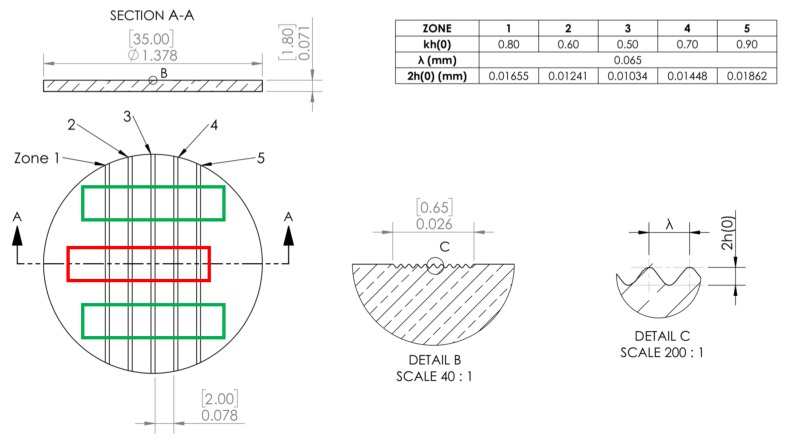
Target geometry with multiple surface perturbations on the copper sample used for the gun-drive experiments. The red region represents the area implanted with helium, whereas the green area shows the region used to extract data corresponding to unimplanted copper. Please note that the dimensions on the actual drawing are in inches with mm in square brackets.

**Figure 2 materials-13-01270-f002:**
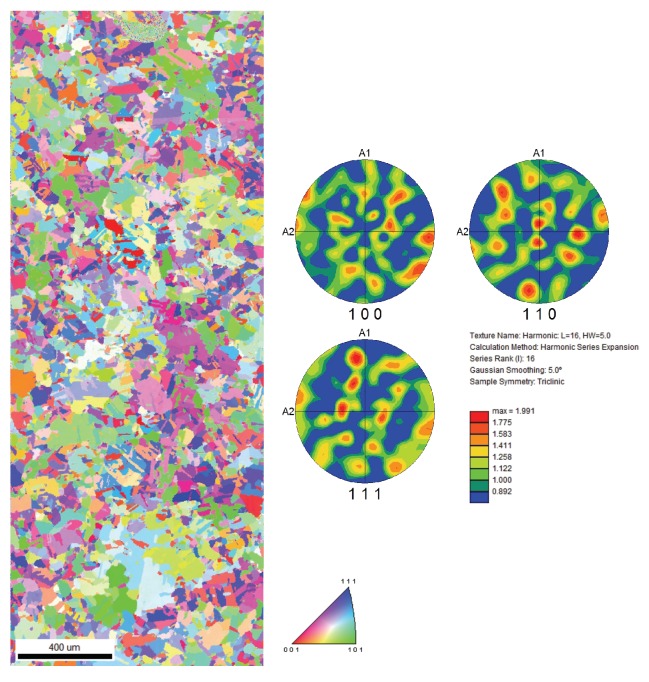
Microstructure of Cu after Helium implantation, characterized using Electron Back Scatter Diffraction along with the texture plot. This shows that the grain size and texture of copper were not altered due to the implantation process.

**Figure 3 materials-13-01270-f003:**
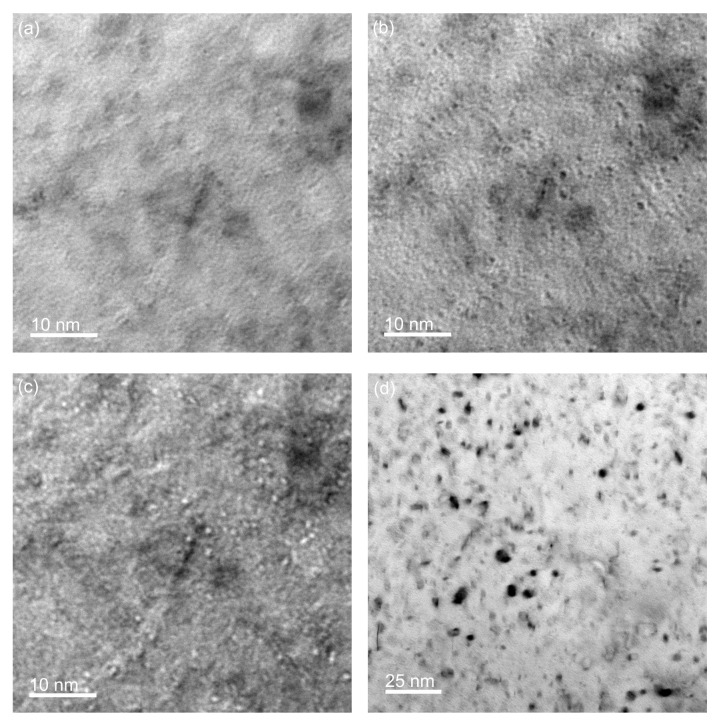
Small helium bubble (2-nm diameter) were observed in the implanted samples using through-focus imaging (±1 μm). Bubbles are nearly invisible in the (**a**) focused image, but appear dark and light in the (**b**) overfocused and (**c**) underfocused conditions, respectively. Additional displacement related defects (e.g., dislocation loops) were observed, as seen in the (**d**) 4000 appm implanted sample.

**Figure 4 materials-13-01270-f004:**
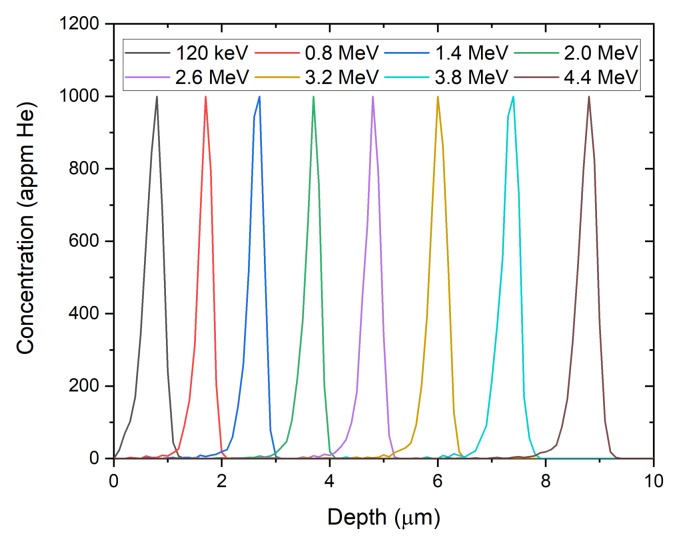
Helium concentration with depth into a copper target, as simulated using SRIM for each energy used here.

**Figure 5 materials-13-01270-f005:**
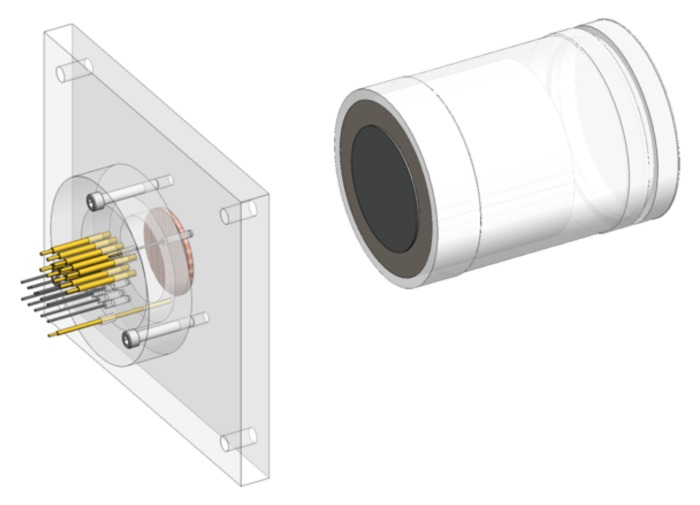
**Left**: Image showing the sample in the lexan plate that is used to mount the target in the gun along and right along with full the diagnostic assembly. **Right**: the projectile with the tantalum impactor.

**Figure 6 materials-13-01270-f006:**
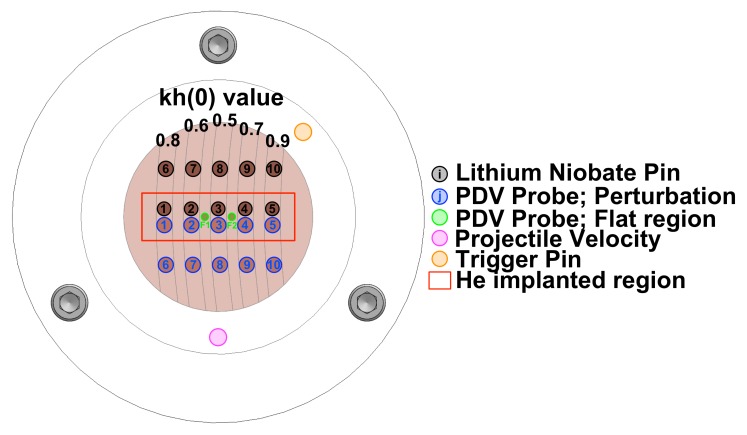
Schematic of the rear of the diagnostics package and target.

**Figure 7 materials-13-01270-f007:**
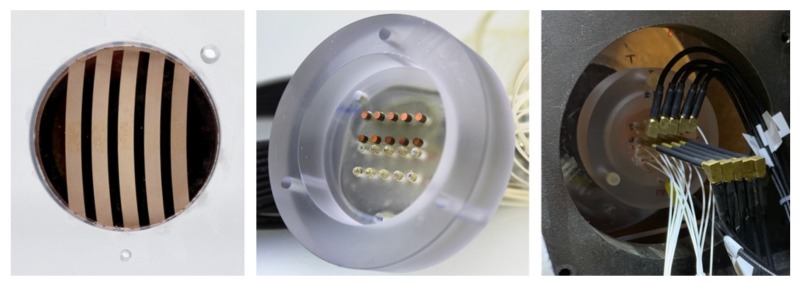
Images showing **left**—rear of the target mounted in the lexan plate, **center**—back of the diagnostics package showing the LN pins (top two rows) and PDV probes (bottom two rows), **right**—rear of the target and diagnostics assembly mounted in the gas gun.

**Figure 8 materials-13-01270-f008:**
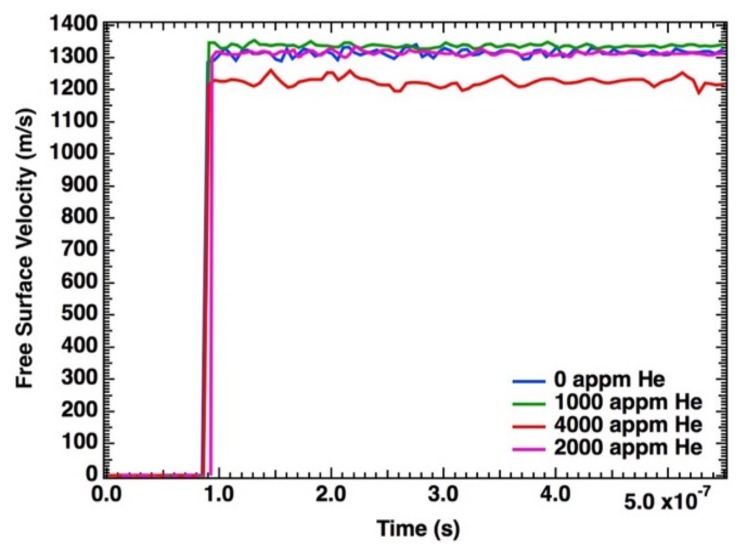
Velocity–time history from one of the flat regions for each of the four experiments.

**Figure 9 materials-13-01270-f009:**
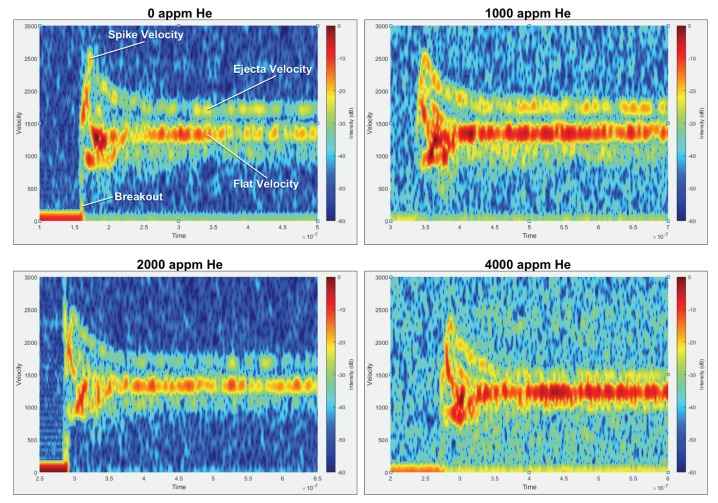
Spectrograms corresponding to copper with (**top-left**) 0 appm, (**top-right**) 1000 appm, (**bottom-left**) 2000 appm, and (**bottom-right**) 4000 appm helium.The spectrograms are colored by intensity, which varies between experiments depending on the signal to noise ratio for that particular experiment.

**Figure 10 materials-13-01270-f010:**
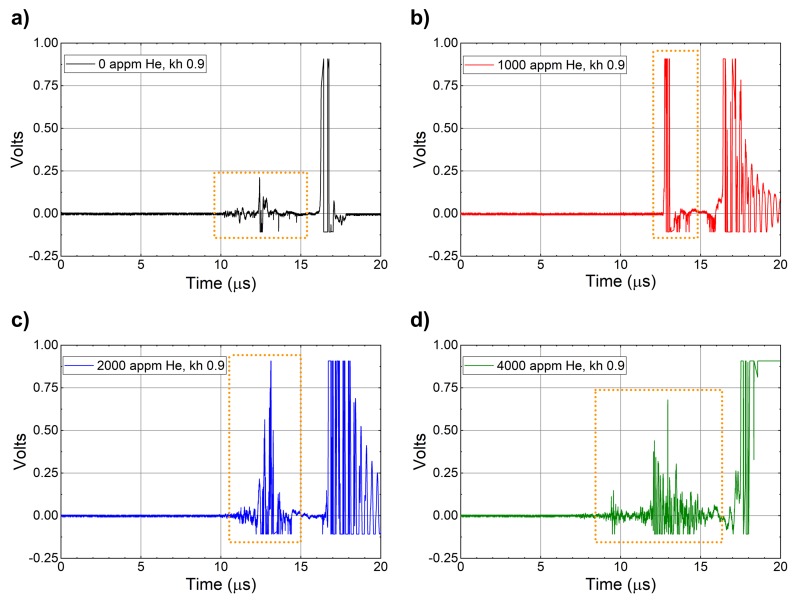
Voltage–time history corresponding to (**a**) 0 ppm, (**b**) 1000 ppm, (**c**) 2000 ppm, and (**d**) 4000 ppm for the four shots obtained from LN pins placed at a kh of 0.9. The dotted boxes outline the beginning and start times associated with ejecta at the LN pins. The late time signal is attributed to the free surface of the copper impacting the LN pins.

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
