# Peer review of "The Role of Helium on Ejecta Production in Copper"

_materials, 2020, doi:10.3390/ma13061270_

Round 1
Reviewer 1 Report
Figure 2: How the TEM sample prepared? If the samples were prepared by FIB, the dark dots in Figure2 look like FIB damages rather than helium bubbles to me. If the authors claim the images were taken by through focus method, please provide under focus, in focus and over focus images.
Bubbles should present bright dots in under-focus and dark dots in over-focus images.
Figure 3: I don't understand how to use EBSD to defect helium. I am sure that the authors made a mistake.
Author Response
Dear Reviewer,
Please find attached a revised copy of the manuscript entitled “The Role of Helium on Ejecta Production in Copper” by Fensin et al. We thank the referees for their comments, all of which have been taken into account in modifying the manuscript. Having attempted to follow all of these recommendations, we are resubmitting our manuscript for further consideration. A summary of the changes made in response to the referee’s comments is given below.
Yours Sincerely,
Saryu Fensin
----------------------------------------------------------------------
Comments
Reviewer 1:
Figure 2: How the TEM sample prepared? If the samples were prepared by FIB, the dark dots in Figure2 look like FIB damages rather than helium bubbles to me. If the authors claim the images were taken by through focus method, please provide under focus, in focus and over focus images.
Reply: Thank you so much for your comment. We apologize for not being clear. We have added the following text in the methodology section of the paper, “ To preserve the reflecting surfaces of the RMI targets, characterization specimens were collected from surrogate copper samples prepared from the same plate and implanted under the same conditions as the targets. Standard metallographic techniques were used to prepare samples for EBSD analysis in a FEI Inspect scanning electron microscope (SEM). A final polish of 1 μm Al2O3 was followed by an electrochemical polish in a 2:1 solution of phosphoric acid and water at 1.9 V for 30-40 sec. The samples were then lightly etched using a solution of FeCl3 and HCl. Surface preparations were completed prior to any implantations. Post-implantation grain sizes (60 μm) and textures (random) were found to be similar to unimplanted copper [12] and are shown in Fig. 2. TEM foils were then extracted from these same surrogates and thinned to electron transparency according to conventional “lift-out” techniques in a FEI Helios 600 focused ion beam (FIB) system. Final polishing was performed to minimize preparation induced damage artifacts, using 2 kV Ga+ ions in the same FIB.”. We have also added the under, in and over focus images in Figure 3.
Bubbles should present bright dots in under-focus and dark dots in over-focus images.
Reply: Please refer to the newly added Figure 3 that has been corrected per your comments to highlight the bubbles correctly.
Figure 3: I don't understand how to use EBSD to defect helium. I am sure that the authors made a mistake.
Reply: We apologize for not being clear. The purpose of the EBSD image was not to characterize the morphology of the helium. But to show that implantation did not alter the microstructure (grain size, texture) of the copper material. This was done such that we could quantitatively compare our results to Copper with no helium. We thought it was important to ensure that the microstructures were the same, the only difference being the presence of Helium (which is not characterized by EBSD but by TEM). We have added the following text in the manuscript to alleviate this concern, “Since we were interested in studying the effect of He on ejecta production in copper, it was essential to verify that the initial microstructure (grain size, texture) for copper was not altered during the implantation process. This was essential to ensure quantitative comparison between data obtained from un-implanted and implanted copper. The change in the overall microstructure was quantified before and after irradiation using electron back-scattered diffraction (EBSD) with a FEI Inspect. The morphology of the implanted helium was investigated through the use of transmission electron microscopy (TEM) with a FEI Tecnai F20.”
Reviewer 2 Report
Work under review is devoted to study the effect of surface and subsurface heterogeneities on shock wave induced ejecta production. Several targets were prepared in order to investigate combination of surface roughness and He implantation on this process. Despite well described experimental setup, results are presented very sparingly, which does not allow a reader to evaluate the whole picture obtained by the authors of the study. For example, in fig. 3 shows the microstructure of only one of the implanted samples and no comparison is made with the original target, or targets implanted to other doses. Despite 5 kh product values are declared, in the Results section data on only one of them are analyzed. Micrographs of the surface after shock-wave impact is also not presented. Thus I suggest authors to improve presentation of data and analysis before acceptance.
Author Response
Dear Reviewer,
Please find attached a revised copy of the manuscript entitled “The Role of Helium on Ejecta Production in Copper” by Fensin et al. We thank the referees for their comments, all of which have been taken into account in modifying the manuscript. Having attempted to follow all of these recommendations, we are resubmitting our manuscript for further consideration. A summary of the changes made in response to the referee’s comments is given below.
Yours Sincerely,
Saryu Fensin
----------------------------------------------------------------------
Comments
Reviewer 1:
Work under review is devoted to study the effect of surface and subsurface heterogeneities on shock wave induced ejecta production. Several targets were prepared in order to investigate combination of surface roughness and He implantation on this process. Despite well described experimental setup, results are presented very sparingly, which does not allow a reader to evaluate the whole picture obtained by the authors of the study. For example, in fig. 3 shows the microstructure of only one of the implanted samples and no comparison is made with the original target, or targets implanted to other doses. Despite 5 kh product values are declared, in the Results section data on only one of them are analyzed. Micrographs of the surface after shock-wave impact is also not presented. Thus I suggest authors to improve presentation of data and analysis before acceptance.
Reply: Thank you for your comments. We have addressed some of your concerns by modifying the manuscript per comments from other reviewers also. However, the reason we only show one image in Fig. 3 is because we referenced our other work that shows the texture and grain morphology for a similar copper and since there isn’t a change we didn’t want to add similar images twice. The reason for discussing just one kh is that only two of them went unstable and produced ejecta. We were not sure in the beginning which kh would actually produce ejecta. To clarify this we have added the following text to the paper: “We decided to focus on discussing results from kh of 0.9 in this paper as the focus of this work is on the effect of helium on ejecta production rather than kh”

Reviewer 3 Report
I have small questions and comments
Report to Crystals.
The authors use different experimental techniques to study the effect of He concentration and morphology on ejecta production. The main results of this work are the increase of 30% in the velocity at which the ejecta cloud was traveling in Copper with 4000 appm as compared to its unirradiated sample. I have small questions regarding their assumptions. However, overall, I believe that this manuscript is a valuable contribution to the literature and, once my comments below are adequately addressed, should be published in Materials.
Comments and suggestions
1. Figure 2 does not contain (a, b, c), please add these.
2. The images show the bubbles embedded in Cu matrix after He irradiation, Even is not the topic of the paper I would see one sentence about the formation of the He bubbles because I think the role of these bubbles in ejecta mass is important.
Bubbles formation: During irradiation, many defects are formed such as interstitials (I), vacancies (V)….
Due to the low electron density around the vacancies, He prefers to trap inside these vacancies and form the vacancy complexes VnHem which is the precursor of the bubbles [i, ii]. Then VnHem grows by the gain of the chemical potential [i].
3. Figure 3 and 5: the size of the letter on the right side is too small please please make it clear.
4. The presence of He increases total ejected mass as compared to pure Cu. I’m a bit surprised that the authors do not explain the increase of ejecta and velocity for high He fluence 4000 appm. Is due to He bubbles?
5. You should speak about the role of He bubbles compared atomic He (when He are in an interstitial site in Cu matrix) I think this increase is attributed to non-planarities in the shock front and reflected pulse due to He bubbles. In addition to this variability, the shock wave compresses the bubbles, causing internal jetting which further contributes to ejected mass from the surface. The formation of small jets suggests that the presence of any heterogeneity with a lower/higher density, compared to the parent metal matrix, leads to the creation of a non-planarity in the shock front whose magnitude is proportional to the difference in density and size of the heterogeneity.
[i] N. Daghbouj , B.S. Li , M. Callisti , H.S. Sen , M. Karlik , T. Polcar , Microstructural evolution of helium-implanted 6H-SiC subjected to different irradiation conditions and annealing temperatures: a multiple characterization study, Acta Materialia 181 (2019) 160-172.
[ii] H.S. Sen, T. Polcar, Vacancy-interface-helium interaction in Zr-Nb multi-layer system: A first-principles study, J. Nucl. Mater. 518 (2019) 11-20.
Author Response
Dear Editor,
Please find attached a revised copy of the manuscript entitled “The Role of Helium on Ejecta Production in Copper” by Fensin et al. We thank the referees for their comments, all of which have been taken into account in modifying the manuscript. Having attempted to follow all of these recommendations, we are resubmitting our manuscript for further consideration. A summary of the changes made in response to the referee’s comments is given below.
Yours Sincerely,
Saryu Fensin
----------------------------------------------------------------------
Comments
Reviewer 1:
Figure 2 does not contain (a, b, c), please add these.
Reply: We have actually moved Fig. 2 to Fig. 3 in the manuscript and added under-, in- and over- focus TEM images to highlight the helium morphology.
The images show the bubbles embedded in Cu matrix after He irradiation, Even is not the topic of the paper I would see one sentence about the formation of the He bubbles because I think the role of these bubbles in ejecta mass is important.
Reply: We have added the following text to the paper: “The formation of helium bubbles in these samples is to be expected, given the favorable trapping of excess point defects at dissolved helium atoms and the suppressive effect of helium on the critical radii for cavity growth [mansur 1986]. The authors acknowledge, however, that helium was not introduced homogeneously by the implantation method used here, instead concentrating in bands around the peak ion ranges of each implantation energy. With the low diffusivity of helium in copper near room temperature, this leads to a non-uniform distribution of helium bubbles with depth. Future studies on the effects of helium distribution will incorporate ion energy degradation techniques to allow for more uniform implantation of helium at elevated temperature”
Figure 3 and 5: the size of the letter on the right side is too small please make it clear.
Reply: We apologize for the inconvenience and have modified the Figures.
The presence of He increases total ejected mass as compared to pure Cu. I’m a bit surprised that the authors do not explain the increase of ejecta and velocity for high He fluence 4000 appm. Is due to He bubbles?
Reply: We did not add a reference to some modeling work where we had discussed hypothesis regarding why this might be happening. We have added the following text to the paper: “This is somewhat consistent with our recent results from molecular dynamics simulations that show a large difference in total ejected mass from copper with and without helium \cite{flanagan}. Through our modeling work, we had hypothesized that the presence of helium bubbles can lead an an increased ejected mass due to two reasons: 1) the bubbles alter the shock velocity in their vicinity due to differences in their density as compared to copper. This leads to the formation of a non-planar shock front that can increase ejecta production and 2) the shock wave can compress the bubble and lea to internal jetting similar to shape-charge phenomenon. So although scientifically an increase in ejecta mass is plausible, we were not able to obtain a quantitative value for the ejected masses from these experiments.”
You should speak about the role of He bubbles compared atomic He (when He are in an interstitial site in Cu matrix) I think this increase is attributed to non-planarities in the shock front and reflected pulse due to He bubbles. In addition to this variability, the shock wave compresses the bubbles, causing internal jetting which further contributes to ejected mass from the surface. The formation of small jets suggests that the presence of any heterogeneity with a lower/higher density, compared to the parent metal matrix, leads to the creation of a non-planarity in the shock front whose magnitude is proportional to the difference in density and size of the heterogeneity.
Reply: Please see the added text above.

Round 2
Reviewer 1 Report
I have no further questions
Reviewer 2 Report
Despite authors have addressed some of my suggestions, improvement of data analysis is still needed.
One of points is that authors state “lithium Niobate pins are not the most reliable diagnostic for measuring solid mass”. Then, I doubt the results regarding material properties and the paper should be submitted to other journal aimed to experimental technique analyses.
The main concern is that I cannot realize the main result of this paper. What do authors wish to tell the reader? What kind of “advance the in-depth understanding of the relationship between the structure, the properties or the functions of all kinds of materials” follows from the manuscript? To my opinion this might be addressed in the Discussion section.
Some minor suggestions as follows:
It would be nice to support speculation concerning the distribution of implanted He in the Cu target with some simulation of the implantation process. Well established TRIM code can give a good insight on this.
Caption to the Fig.1 misprint: “The red region represents the area implanted with copper…”, replace “copper” with “helium”
Caption to the Fig.3 describes 4 images, whereas only 3 were placed
Conclusions section is too long. I suggest include Discussion section and shift part of the text into it.
Author Response
Dear Editor,
Please find attached a revised copy of the manuscript entitled “The Role of Helium on Ejecta Production in Copper” by Fensin et al. We thank the referees for their comments, all of which have been taken into account in modifying the manuscript. Having attempted to follow all of these recommendations, we are resubmitting our manuscript for further consideration. A summary of the changes made in response to the referee’s comments is given below.
Yours Sincerely,
Saryu Fensin
----------------------------------------------------------------------
Comments
Reviewer 2 (Round 2):
Despite authors have addressed some of my suggestions, improvement of data analysis is still needed.
One of points is that authors state “lithium Niobate pins are not the most reliable diagnostic for measuring solid mass”. Then, I doubt the results regarding material properties and the paper should be submitted to other journal aimed to experimental technique analyses.
Reply: The reason we added the comment about the lithium Niobate pins is because we were not able to get a “quantitative mass from the diagnostics”. However, using the pins we were able to show two things 1) Ejecta was produced for a longer time as we added more helium into the material suggesting an increase in mass and 2) An increase in helium concentration lead to an increase in the velocity at which the ejecta cloud is traveling. The focus of the paper is still answering the question how does helium concentration affect ejecta production in Copper. We have reduced the emphasis on this point and removed it from conclusions.
The main concern is that I cannot realize the main result of this paper. What do authors wish to tell the reader? What kind of “advance the in-depth understanding of the relationship between the structure, the properties or the functions of all kinds of materials” follows from the manuscript? To my opinion this might be addressed in the Discussion section.
Reply: We are sorry that the main result of the paper is not clear. To summarize the main points, we want to make are: 1) Increasing the Helium concentration seems to increase the total ejected mass and 2) Increasing the helium concentration increases the velocity at which ejecta is traveling. This is related to the fact that as we are increasing the helium concentration, we are forming small Helium bubbles that 1) alter the morphology of the shock wave and 2) burst under shock adding more kinetic energy into the system leading to an increase in the velocity of the ejecta cloud. We are in essence altering the structure of the material by addition of helium to the surface which is in-turn altering ejecta production from the surface of the material. This is significant because this has never been shown before. It is believed that heterogeneities in the material should not alter ejecta production significantly. We show in this work that this statement is not true. We have added text to this affect in the conclusions.
Some minor suggestions as follows:
It would be nice to support speculation concerning the distribution of implanted He in the Cu target with some simulation of the implantation process. Well established TRIM code can give a good insight on this.
Reply: We have Figure 4 with this information in the methodology section.
Caption to the Fig.1 misprint: “The red region represents the area implanted with copper…”, replace “copper” with “helium”
Reply: We apologize for the typo and have fixed it.
Caption to the Fig.3 describes 4 images, whereas only 3 were placed
Reply: There are four images in Fig 3 labeled a to d. I am not sure why they weren’t showing up in your version. Can you please look at this newer version.
Conclusions section is too long. I suggest include Discussion section and shift part of the text into it.
Reply: We have taken your suggestion and shortened the conclusion section to the following: “The goal of this work was to understand the effect of heterogeneities like helium on ejecta production in metals. OFHC copper was used as a model material for this study. Gas-gun driven RMI experiments, which required special perturbations be machined onto the copper targets, were used to investigate the effect of helium concentration on ejecta production. Four different concentrations of helium, 0, 1000, 2000 and 4000 appm were implanted into the copper sample at the Michigan Ion Beam Laboratory.
These samples were then subjected to shock loading using a gas-gun and diagnostics were used to measure the velocity-time history and mass-time history as the perturbations inverted, grew and eventually went unstable to produce ejecta. Our results show that kh of 0.8 and 0.9 produced ejecta in all targets regardless of the presence of helium. Further analysis of the velocity and mass data associated with kh of 0.9 showed 1) an increase in the production of finer ejecta traveling at higher velocities as a function of helium concentration and 2) longer times associated with a finite voltage suggesting an increase in mass as a function of Helium concentration. These results are significant and show that heterogeneities like Helium can actually alter important dynamic properties like ejecta production from metals. “